# Does Antenatal Lactoferrin Protect Hippocampal Development in Ovine Fetuses with Growth Restriction?

**DOI:** 10.3390/cells14241951

**Published:** 2025-12-09

**Authors:** Dahyun Kang, Ingrid Dudink, Tegan A. White, Amy E. Sutherland, Tamara Yawno, Yen Pham, Petra S. Huppi, Stéphane V. Sizonenko, Suzanne L. Miller, Beth J. Allison

**Affiliations:** 1The Ritchie Centre, Hudson Institute of Medical Research, Clayton, VIC 3168, Australia; 2Department of Obstetrics and Gynaecology, Monash University, Clayton, VIC 3168, Australia; 3Department of Paediatrics, Monash University, Clayton, VIC 3168, Australia; 4Department of Pediatrics, Gynecology and Obstetrics University of Geneva, 1205 Geneva, Switzerland

**Keywords:** fetal growth restriction, preterm, brain injury, brain pathology, lactoferrin, neuroprotection, cardiovascular

## Abstract

Early-onset fetal growth restriction (FGR) is associated with prolonged fetoplacental hypoxia and altered brain development, including deficits in hippocampal structure and function. Neuroprotective actions of lactoferrin have been described, mediated via anti-inflammatory and antioxidant properties. Here, we investigated whether the antenatal administration of lactoferrin (1) improves hippocampal structure, (2) promotes neuronal growth, and (3) mitigates neuroinflammation in the hippocampus of fetal sheep with FGR. Early-onset FGR was induced by performing single umbilical artery ligation surgery on ovine fetuses at ~89 days gestational age (dGA; term ~148 dGA), compared with appropriate for gestational age (AGA) controls. Lactoferrin supplementation to the ewe commenced at 95 dGA (oral, 36 g/day) and continued until 127 dGA (fetal group) or birth (newborn group). Experimental fetal groups included control appropriate for gestational age (AGA; n = 8), FGR (n = 5), control + lactoferrin (AGA + Lacto; n = 6), and FGR + lactoferrin (FGR + Lacto; n = 6). In the fetal group, results showed that neither FGR nor lactoferrin altered hippocampal structure at 127 dGA. Lactoferrin exposure significantly increased neuronal abundance but also altered neuronal morphology. Lactoferrin increased the neurotrophic factor, brain-derived neurotrophic factor (BDNF) in the hippocampus. Lactoferrin exerted region-specific anti-inflammatory effects, with reduced total microglial cell count and resting microglia count in the *Cornu Ammonis* (CA)3 region only. In the newborn cohort, we observed increased circulating haematocrit concentration in early life. These findings support that antenatal lactoferrin has an anti-inflammatory effect in the fetal brain and increases fetal brain neurotrophic factor BDNF. Still, prolonged exposure during pregnancy may yield mixed effects on fetal brain development and haematological balance.

## 1. Introduction

Placental transfer of oxygen and nutrients to the developing fetus during pregnancy is critical, and placental dysfunction adversely impacts fetal growth. Complications of placental development and function, usually combined under the umbrella term placental insufficiency, are the primary cause of fetal growth restriction (FGR) [1]. FGR is a pathological stunting or reduced trajectory of fetal growth in utero, predominantly due to a suboptimal supply of oxygen and nutrients [2]. Placental insufficiency and FGR significantly increase the risk of perinatal mortality and contribute to morbidities of respiratory, cardiovascular, and metabolic function [3,4,5,6,7]. These morbidities arise from disrupted organ development after fetoplacental hypoxia, inflammation, and oxidative stress, which affect major organs, including the lungs, heart, adrenals, and brain [3,8] and the comorbidity of preterm birth [9]. Fetal brain development may be particularly sensitive to placental compromise, with both clinical imaging and preclinical studies demonstrating subtle but distinct neuropathology associated with FGR that is first evident antenatally [10,11,12], underscoring the antenatal period as a potential therapeutic window for the prevention of altered brain development.

The hippocampus is a critical neuron-rich brain region that regulates learning and memory processing functions and has been shown to be highly susceptible to placental insufficiency [5,13]. The hippocampus undergoes a period of rapid neural growth during the third trimester of human pregnancy, which involves neurite sprouting, dendritic arborisation, and synaptogenesis [13,14]. This critical developmental window often coincides with the timing of chronic hypoxic compromise in pregnancies diagnosed with early-onset FGR. In early-onset FGR, hippocampal volume is reduced [15,16,17,18,19], with persistent cognitive, memory, and behavioural deficits [20,21,22]. In the current study, we concentrated on hippocampal neurodevelopmental deficits associated with FGR and assessed whether antenatal treatment with lactoferrin was neuroprotective. Sheep offer a practical and biologically relevant model for prenatal brain development because their gestation length and trajectory of neural maturation are far closer to humans than those of small-animal models. Their prolonged, longitudinal pattern of brain growth mirrors key human processes, including the extended maturation of the hippocampus and the parallel demands placed on placental physiology during late gestation. Although the sheep brain is slightly more mature at term than the human newborn brain, the moderately preterm stage used in this study is well matched to late-preterm human brain development. Given the hippocampus was our primary region of interest, and its dendritic growth peaks late in gestation and extends into early life, the developmental timing in sheep provides an advantageous window for examining these protracted neural processes.

Lactoferrin, an iron-binding glycoprotein predominantly found in breast milk [23], may be a promising neuroprotective candidate for FGR. It crosses the blood–brain barrier via receptor-mediated transcytosis [24] and exerts multifaceted actions, including anti-inflammatory and antioxidant effects [25,26]. Previous studies have reported that lactoferrin inhibits key inflammatory pathways, such as NF-κB signalling [27,28], and downregulates pro-inflammatory cytokines in rodent models of pregnancy compromise [29,30,31,32]. Lactoferrin demonstrates antioxidant effects in the brain, as evidenced by the reduced production of reactive oxygen species (ROS) [30,33,34] and upregulation of antioxidant proteins in young rodents [27,29]. These combined anti-inflammatory and antioxidant effects underlie lactoferrin’s neuroprotective effects in the hippocampus, including the attenuation of neuronal damage in mice [28], preservation of neuronal growth in piglets [35], and restoration of hippocampal volume loss in rats [36]. Lactoferrin supplementation exhibits a low toxicity profile and has received FDA approval [37,38], supporting its translatability.

The *antenatal* tneuroprotective efficacy of lactoferrin remains largely unknown, particularly when administered chronically, and has never been studied in a long-term large-animal model. This study aimed to determine whether lactoferrin supplementation to the mother for a period between approximately 28 to 34 weeks of human gestation (sheep gestational age range 95 days to 127 days, where term is ~148 days) was neuroprotective for hippocampal development within the brain of FGR fetuses. We examined two subregions of the hippocampus, the CA1 and CA3 areas, which are critical for hippocampal memory and executive processing functions, but also vulnerable to perinatal compromise [13,39]. The study was conducted across two cohorts with four treatment groups of interest: control (appropriately grown for gestational age, AGA), growth-restricted (FGR), lactoferrin-treated control (AGA + Lacto), and lactoferrin-treated (FGR + Lacto). We hypothesised that maternal administration of lactoferrin during gestation would be neuroprotective in the hippocampus, via anti-inflammatory and antioxidant actions, promoting neuronal growth in the CA1 and CA3 hippocampal regions.

## 2. Materials and Methods

### 2.1. Ethics Approval

All animal experiments were approved by the Monash Medical Centre Animal Ethics Committee (MMCA 2016/62, MMCA 2021/19) and were performed in accordance with the National Health and Medical Research Council (NHMRC) of Australia and the Australian Code for the Care and Use of Animals for Scientific Purposes. All experimental design and animal procedures are reported in accordance with the ARRIVE guidelines, with a completed checklist provided as Appendix A.

### 2.2. Surgery to Induce FGR

Time-mated twin-bearing mixed-breed pregnant ewes (*Ovis aries*) underwent single umbilical artery ligation (SUAL) surgery between 88 and 90 days of gestation (dGA; 0.6 gestation; term ~148 days) to induce placental insufficiency and replicate early-onset human FGR, as previously described [8,11,40]. Briefly, we induced general anaesthesia (1–2.5% isoflurane in oxygen, Bomac, Hornsby, NSW, Australia), and the lower body of each fetus was removed from the uterus in turn, with SUAL undertaken in one fetus, while the umbilical cord of the twin AGA fetus was handled but not ligated. A catheter was inserted into the maternal jugular vein for the administration of post-operative antibiotics and euthanasia at the completion of the experiment. At the cessation of the surgical procedure and after withdrawal of anaesthesia, the ewe was monitored closely and given oral Panadol (1 g), and intravenous ampicillin (1 g) and engemycin (500 mg), for three days. Ewes were housed together in individual pens in a 12 h light–dark cycle with free access to food and water and monitored daily.

### 2.3. Bovine Lactoferrin Supplementation

Ewes were randomly allocated to receive either oral administration of water (AGA, n = 8; FGR, n = 5) or lactoferrin (AGA + Lacto, n = 6; FGR + Lacto, n = 6). Lactoferrin was prepared as a mixture containing 36 g of bovine-derived lactoferrin powder (~0.5 g/kg/d; 95% apo-lactoferrin, NutriScience Innovations, LLC, Milford, CT, USA; native-lactoferrin, Saputo, Melbourne, VIC, Australia, average maternal weight ~59 kg [40]), 5 mL of 12.5% glycerine, and ~70 mL of water. The lactoferrin dose was adapted from a study conducted in rodents [31] and scaled up for pregnant sheep. All ewes readily consumed the lactoferrin-supplemented feed, and no adverse effects or changes in general health were observed. Control ewes received water only. In an earlier pilot study, a cohort of ewes were treated with the glycerine vehicle only, and we detected no differences in hippocampal width, NeuN abundance, or Iba1+ve cell abundance; we therefore did not pursue a full group of vehicle animals, and while this is a study limitation, it is unlikely to influence the study outcomes.

In the fetal study, lactoferrin was administered orally to ewes daily from 95 dGA until postmortem at 127 dGA, when ewes and fetuses were euthanised via intravenous pentobarbitone (100 mg/kg; Virbac Pty Ltd., Milperra, Australia). In the newborn group, FGR, AGA, and FGR + Lacto animals underwent this same procedure, but the lactoferrin (or water) was continued daily until birth was induced at 136 dGA [40]. To induce preterm birth, ewes received mifepristone (50 mg; Linepharma, MSHealth, Richmond, VIC, Australia) and betamethasone (11.4 mg; Celestone Chronodose, Schering-Plough, North Ryde, NSW, Australia) intramuscularly at 48 h before birth (134 dGA), and a second dose of betamethasone (11.4 mg) 24 h before birth. We monitored for signs of labour remotely via CCTV, and an experimenter was present at the time of birth for all lambs. Lambs were kept with their mothers until postmortem at either 24 h after birth or 4 weeks postnatal age, when animals were euthanised as above.

### 2.4. Brain Processing and Immunohistochemistry

At the scheduled postmortem, the fetus was weighed and the fetal or neonatal brain was removed, weighed, and divided sagitally into left and right hemispheres. The right hemisphere was sectioned coronally into 5 mm slices and immersion-fixed in 10% Neutral Buffered Formalin (Trajan Scientific, Ringwood, Australia) for 5 days. This was followed by standard paraffin embedding for immunohistochemistry using standard protocols, prior to sectioning with a microtome at a thickness of 10 µm at the mid-thalamic level, containing the hippocampus (corresponding to Section 1120 in the Sheep Ovis Aries atlas, Michigan State University). Sections were mounted on SuperFrost Plus slides (Thermoscientific, Scoresby, VIC, Australia). The left hemisphere was divided into anatomical regions and snap-frozen for additional analysis if required. All immunohistochemical procedures described hereafter were undertaken only in the fetal (127 dGA) cohort. Duplicate sections containing the CA1 and CA3 hippocampal regions were stained with neuronal nuclei (NeuN), ionised calcium-binding adaptor molecule 1 (Iba-1), microtubule-associated protein 2 (MAP2), and BDNF [11,41,42] (Appendix A). Positive and negative control sections were examined. For all immunohistochemical and immunofluorescence analyses, experimental conditions were standardised within each assessment: tissues were collected at the same developmental stage, fixed and processed using identical protocols, and stained with the same antibody in the same run. Negative controls and blocking peptides were included to confirm specificity.

### 2.5. Image Analysis

The CA1 and CA3 regions of the hippocampus were imaged at 30× magnification (Olympus BX-41, Tokyo, Japan, and Olympus BX-53, Tokyo, Japan) for sections stained with NeuN and Iba-1. Positively stained cells were manually counted and sub-categorised using QuPath (University of Edinburgh, UK, version 0.5.1). NeuN has traditionally been used to quantify mature neurons, but it has also been identified as the pre-mRNA alternative splicing protein Rbfox3 [43]. Consequently, this study employed the profile of NeuN/Rbfox3 immunostaining to sub-categorise neuronal morphology for the first time in the fetal sheep hippocampus, using the staining profiles described recently for neonatal piglets following a hypoxic–ischaemic insult [44]. As per Primiani et al., the neurons were classified into four phenotypes: (1) healthy neurons (Figure 2E), (2) Rbfox3-depleted neurons (Figure 2H), (3) degenerating neurons (Figure 2K), and (4) abnormal nuclear chromatin neurons (Figure 2N). Healthy neurons were morphologically intact with well-defined NeuN/Rbfox3 staining. Rbfox3-depleted neurons exhibited a faded appearance, with a defined cell membrane, indicating protein depletion. Degenerating neurons demonstrated cytoplasmic shrinkage and a clear perinuclear halo with pyknotic nuclei. Abnormal nuclear chromatin neurons displayed intensely stained Rbfox3 clumping with a white cytoplasmic background. The number of neurons for each phenotype was manually counted on coded slides and presented as the number of cells per mm^2^.

Microglial activation was quantified following previously established criteria [45,46] as (1) resting (Figure 4E), (2) intermediate (Figure 4H), or (3) activated (Figure 4K). Hippocampal structural changes were assessed by averaging 10 measurements of the width of the individual stratum layers, including the stratum oriens (SO), stratum pyramidal (SP), and combined stratum radiatum (SR) and stratum lacunosum moleculare (SL-M) (as illustrated in Figure 1I).

Images from BDNF-stained hippocampal sections were captured at ×200 magnification (Olympus BX-41, Japan). Positive immunofluorescent staining for the densitometric % area of BDNF was acquired using ImageJ v.2.1.0, after setting a background threshold for all images. The density of MAP2 staining within the CA1 and CA3 regions of the hippocampus was assessed on scanned images (Olympus VS120, Monash Histology Platform, VIC, Australia) using Olympus Viewer (OlyVIA ver. 2.9.1). All procedures were performed in a blinded manner by two observers (D.K. and T.A.W.).

### 2.6. Statistical Analysis

Data were averaged between replicated slides and averaged again across groups. The data were then assessed for normality and statistically analysed using two-way ANOVA (*p* < 0.05), evaluating and comparing the effects of growth and treatment, followed by Tukey’s multiple comparisons test, which was applied only when a significant interaction effect was detected. All statistical analyses were performed using GraphPad Prism 10 (version 10.2.1, GraphPad Software, CA, USA) and are presented with individual data points and the mean ± standard error of the mean (SEM).

## 3. Results

### 3.1. Asymmetric Fetal Growth Restriction, Not Modified by Lactoferrin

Growth-restricted fetuses had significantly lower body weights (*p* = 0.002, Table 1), not modified by lactoferrin treatment (*p* = 0.101). Brain weight was not different across groups (*p* > 0.05, Table 1) and, as such, the brain-to-body weight and brain-to-liver ratio were significantly elevated in the FGR cohorts (*p* = 0.003, *p* = 0.032, respectively, Table 1), indicative of asymmetric fetal growth restriction. Lactoferrin treatment did not alter brain sparing (*p* = 0.072) or liver sparing (*p* = 0.471).

### 3.2. Hippocampal Structure Is Preserved

Hippocampal structure was assessed by measuring the width of each stratum layer. Both the CA1 and CA3 regions showed no differences in total width across groups (*p* > 0.05; Figure 1A,B). The individual widths of the stratum oriens (SO), stratum pyramidale (SP), and combined stratum radiatum (SR) and stratum lacunosum moleculare (SL-M) layers were then measured. They were not different across groups in the CA1 and CA3 areas (*p* > 0.05, Figure 1C–H), supporting that no significant differences in hippocampal layer width were detected by FGR or lactoferrin treatment.

### 3.3. Lactoferrin Demonstrates Mixed Effects on Neurons

Using NeuN/Rbfox3 immunohistochemistry, neuronal counts and semi-quantitative morphology characterisation based on cellular appearance were evaluated in the CA1 and CA3 regions. In the AGA brains, NeuN/Rbfox3-positive localisation was nuclear and cytoplasmic (Figure 2E). While FGR animals had a similar neuronal cell count compared to AGA cohorts in both CA1 and CA3 (*p* > 0.05), lactoferrin significantly promoted total neuronal abundance in CA1 (*p* = 0.022, Figure 2A and Figure 5A–D) and more markedly in the CA3 (*p* < 0.0001, Figure 2B) region. The number of each neuronal phenotype was also manually counted. Two-way ANOVA exhibited mixed effects in a region-dependent manner. Semi-quantification revealed that healthy neuron abundance was similar across groups in both the CA1 and CA3 regions (*p* > 0.05, Figure 2C,D). Neither FGR nor lactoferrin altered the Rbfox3-depleted neuronal population in CA1 (*p* > 0.05, Figure 2F). However, lactoferrin was associated with a significant increase in the respective neuronal population in CA3 (*p* < 0.0001, Figure 2G). The number of degenerating neurons remained unaffected by growth or treatment condition in both the CA1 and CA3 regions (*p* > 0.05, Figure 2I,J). Neurons with abnormal chromatin showed a region-dependent response to FGR. They were decreased in CA3 regardless of treatment condition (*p* = 0.021, Figure 2M). Lactoferrin increased the number of neurons with abnormal chromatin in the CA1 (*p* = 0.016, Figure 2L) and CA3 (*p* < 0.0001, Figure 2M) regions.

We assessed the density of BDNF expression, an essential growth factor regulating neuronal survival and development [47] in a semi-quantitative manner. BDNF density was not different between the AGA and FGR cohorts in the CA1 and CA3 regions. In contrast, lactoferrin significantly increased BDNF expression in the CA1 (*p* = 0.0007, Figure 3A and Figure 5E–H) and CA3 (*p* = 0.0002, Figure 3B) regions. MAP2 was assessed as a marker of neuronal and dendrite structure; in the CA3 region, lactoferrin administration resulted in a significant reduction in MAP2 abundance compared to vehicle-treated fetuses (*p* < 0.020, Figure 3D); no other significant treatment effects were observed in the CA3 region. In the CA1 region of the hippocampus, there were no differences in MAP2 abundance (Figure 3C and Figure 5I–L).

**Figure 2 cells-14-01951-f002:**
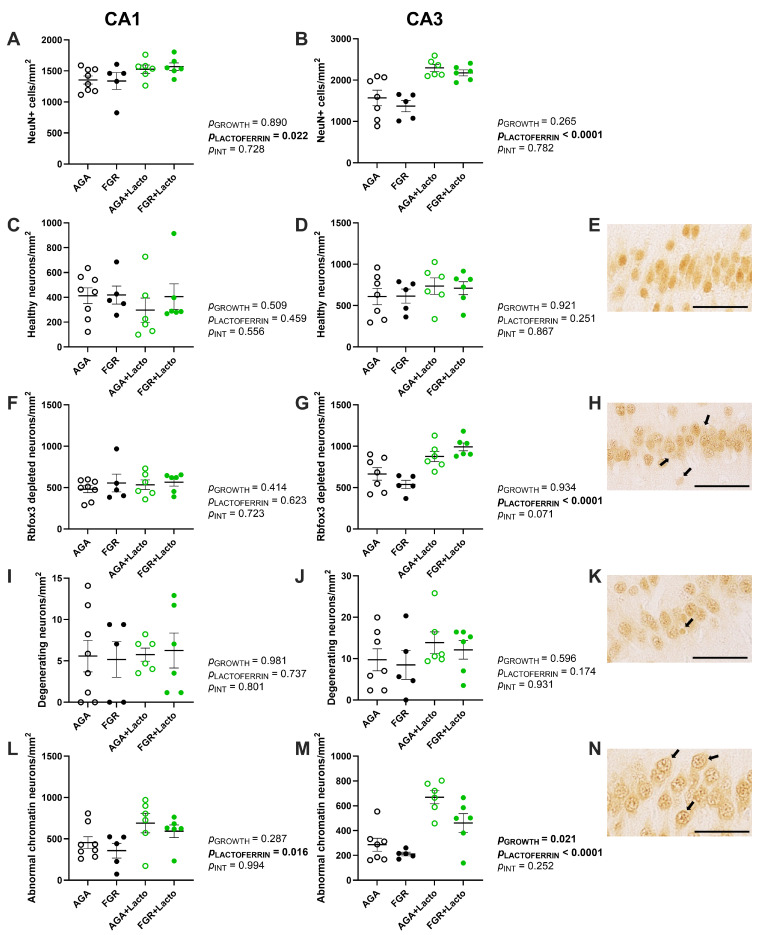
Neuron analyses. Neuronal abundance in CA1 and CA3 (**A**,**B**) and neurons exhibiting each phenotype with example images (taken from the CA1 region (**E**,**H**,**K**,**N**)), as indicated by the arrows—healthy morphology (**C**–**E**), Rbfox3 depletion (**F**–**H**), degenerating (**I**–**K**) and abnormal chromatin (**L**–**N**) were expressed as cells per mm^2^ of the field of view. Groups are AGA (black non-filled, n = 8), FGR (black, n = 5), AGA + Lacto (green non-filled, n = 6), and FGR + Lacto (green filled, n = 6), presented as individual data points and mean ± SEM. Data were statistically analysed by two-way ANOVA, with Tukey’s post hoc, *p* < 0.05. Scale bar = 50 µm.

**Figure 3 cells-14-01951-f003:**
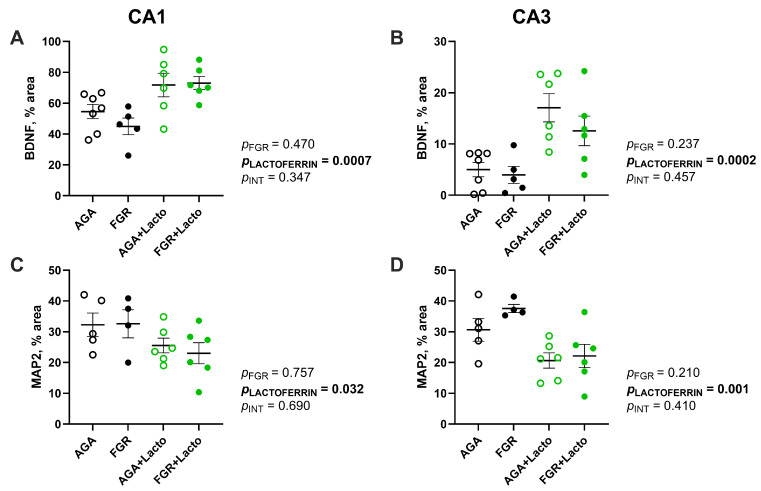
BDNF and MAP2 expression. The % area of BDNF in the CA1 (**A**) and CA3 (**B**) regions was determined. The % area of MAP2 staining was determined in the CA1 (**C**) and CA3 (**D**) regions. Groups are AGA (black non-filled, n = 8), FGR (black, n = 5), AGA + Lacto (green non-filled, n = 6), and FGR + Lacto (green filled, n = 6), displayed as individual data points with lines indicating the mean ± SEM. Data were analysed by two-way ANOVA, with Tukey’s post hoc, *p* < 0.05.

### 3.4. Lactoferrin Alters Microglial Profile in a Region-Dependent Manner

Finally, in the fetal cohort, the total number and activation status of microglia were semi-quantified to examine the inflammatory profile within the hippocampal CA1 and CA3 regions. We observed that FGR did not affect microglial count or morphology in the CA1 region (Figure 4A,C,F,I and Figure 5M–P), and lactoferrin treatment also had no impact in the CA1 region. By contrast, FGR significantly reduced the number of intermediate microglia in the CA3 region (*p* = 0.045, Figure 4G). Lactoferrin did not change this outcome (*p* > 0.05); instead, it decreased the overall microglial count (*p* = 0.041, Figure 4B) and resting microglial count (*p* = 0.028, Figure 4D) in the CA3 region.

**Figure 4 cells-14-01951-f004:**
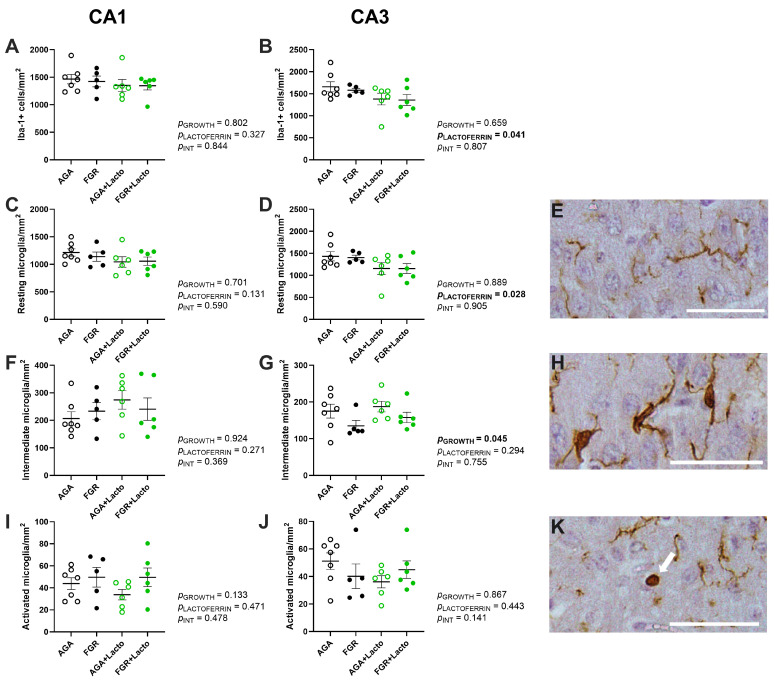
Microglial cell count and activation profile. Microglial abundance (**A**,**B**) and activation states of microglia in the CA1 and CA3 regions were examined and presented as cells per mm^2^. Activation state was classified as resting (**C**,**D**), intermediate (**F**,**G**), and activated (**I**,**J**) phenotypes. Groups are AGA (black non-filled, n = 8), FGR (black, n = 5), AGA + Lacto (green non-filled, n = 6), and FGR + Lacto (green filled, n = 6), displayed as individual data points with lines indicating mean ± SEM. All data were analysed using two-way ANOVA, with Tukey’s post hoc, *p* < 0.05. Representative images from CA1 illustrate resting microglia with a ramified structure (**E**), intermediate microglia with enlarged cell bodies and fewer processes (**H**), and activated microglia with amoeboid morphology as denoted with white arrow (**K**). Scale bar = 50 μm.

**Figure 5 cells-14-01951-f005:**
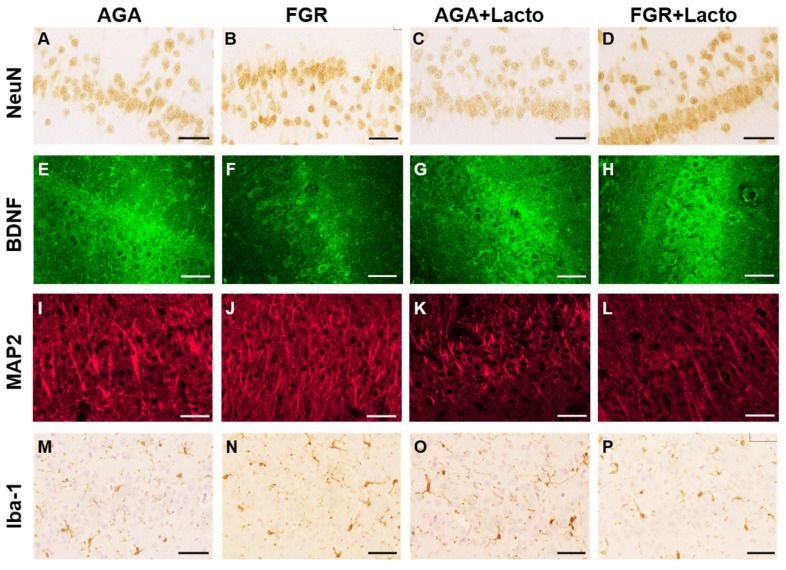
Representative images. Photomicrographs from CA1 that are representative of the group mean data for NeuN (**A**–**D**), BDNF (**E**–**H**), MAP2 (**I**–**L**) and Iba-1 (**M**–**P**) staining. Scale bar = 50 μm.

### 3.5. The Newborn Cohort

The fetal cohort described above indicates a neuroprotective potential for lactoferrin, and accordingly, we aimed to determine whether this was associated with improved postnatal brain function. At 24 h of age, body weight was significantly reduced (*p* = 0.002), and the brain-to-body and liver-to-body weight ratios were significantly increased in FGR lambs compared to AGA (*p* < 0.05, Table 2). Newborn lambs exposed to antenatal lactoferrin were challenging to maintain. Four treated lambs died within 24 h of birth, with postmortem examination revealing markedly elevated haematocrit (~50%), circulatory overload, cardiac hyperplasia, and pulmonary congestion, consistent with an inability to sustain adequate oxygenation. Among the survivors, at 24 h, body weight remained significantly reduced in the FGR + Lacto lambs (*p* = 0.006, Table 2), although the brain-, heart-, liver-, and lung-to-body weight ratios did not differ from AGA. FGR lambs remained significantly lighter than AGA counterparts at 4 weeks, and the FGR + Lacto lambs, all of whom were female, also remained significantly lighter (*p* = 0.007, Table 2) with a significantly increased brain-to-body weight ratio (*p* = 0.033, Table 2) compared to AGA lambs and were not different to FGR untreated lambs.

## 4. Discussion

The developmental trajectory of hippocampal growth is greatest in the final trimester of human pregnancy into the early neonatal period [48], and previous clinical work has shown that FGR is a risk factor for reduced hippocampal volume that can be detected soon after birth [16]. Preclinical studies show that neurodevelopmental deficits associated with FGR are predominantly mediated by a pro-inflammatory and oxidative stress environment that is a consequence of placental dysfunction and chronic fetal hypoxia [8,11,49]. Accordingly, we hypothesised that hippocampal neuropathology in preterm-equivalent fetal sheep with FGR would be protected by antenatal lactoferrin treatment. Contrary to our hypothesis, and despite a significant degree of growth restriction and brain sparing, the FGR cohort did not show notable hippocampal injury. Notwithstanding a lack of pronounced neuropathology in the FGR group, we did observe interesting properties of antenatal lactoferrin treatment that have neuroprotective potential, namely that lactoferrin was associated with a strong anti-inflammatory response within the brain, and lactoferrin increased hippocampal BDNF expression. When lactoferrin was administered over a prolonged (>40 days) period in pregnancy, we found that neonatal circulating haematocrit levels were increased, which is concerning in the setting of chronic fetal hypoxia and circulatory adaptations that already characterise an FGR pregnancy. Combined, these findings suggest that chronic lactoferrin treatment during pregnancy is not uniformly protective, and prolonged antenatal administration is not recommended. Thus, while lactoferrin mediates some protective benefits on the developing brain, an appropriate and optimised application of lactoferrin needs further consideration prior to clinical translation.

We observed two marked effects of antenatal lactoferrin supplementation on the developing fetal brain that are likely to elicit neuroprotective benefits in the setting of perinatal brain compromise. Antenatal lactoferrin treatment induced a significant increase in hippocampal BDNF density across both the CA1 and CA3 subregions, and lactoferrin significantly reduced microglial cell number and shifted the inflammatory profile of microglial cell phenotype within the CA3 region. These are promising actions of lactoferrin that could be harnessed for their protective benefits, with the hippocampus a key brain target. The high rate of hippocampal neurodevelopment in the last month of fetal sheep brain development, corresponding to the final trimester of human pregnancy [8], is one of the critical factors that make the hippocampus susceptible to in utero compromise [13,39]. It was therefore surprising that we did not observe overt hippocampal injury in the FGR sheep fetuses at 127 days of gestation, when, in previous studies, we have detected both white and grey matter (including hippocampal) injury, inflammation, and oxidative stress [8,11,49]. The absence of hippocampal pathology in the current study may reflect the duration of placental insufficiency, species-specific resilience of the ovine hippocampus, or the timing of tissue collection at late-preterm gestation. Additionally, differences in experimental design parameters, such as the severity and timing of FGR induction, may account for these discrepancies. The lack of pathology in the FGR animals has meant that we cannot deduce true neuroprotective actions of lactoferrin; however, there were lactoferrin-induced changes that we would expect to be protective in the setting of perinatal brain injury.

BDNF is a critical neurotrophic factor in the developing brain that is a potent regulator of neuronal development and plasticity, dendrite outgrowth, synaptogenesis, and circuit formation [35,50,51,52]. Preclinical studies show that BDNF gene and/or protein expression is adversely affected by perinatal compromise [53], with guinea pig and fetal sheep studies showing that both acute and chronic hypoxic stress in utero decreases brain, and specifically hippocampal BDNF, expression, and also modifies its high affinity for tyrosine kinase receptor (TrkB) [8,52,54]. BDNF and TrkB expression may be differentially expressed in FGR [52], and we were not able to examine TrkB in the current study, which we acknowledge is a limitation for the interpretation of BDNF. BDNF gene expression remains significantly downregulated at postnatal day 7 in dexamethasone-induced FGR rat pups [53]. In the current study, the potential of lactoferrin to enhance BDNF expression in the CA1 and CA3 hippocampal regions was strong and present in both AGA and FGR fetal brains. Our findings confirm prior preclinical studies showing that maternal lactoferrin supplementation increases BDNF in the hippocampus of young rats that were FGR at birth [53] and postnatal administration from day 3 to 38 in piglets upregulates BDNF gene and protein levels in the hippocampus [35]. The piglet study by Chen et al. demonstrated that the lactoferrin-induced increase in brain BDNF is mediated via transcriptional and posttranslational levels of BDNF in the hippocampus and its signalling pathway, including the downstream target of the BDNF signalling pathway, CREB [35]. Together, these data indicate that in perinatal conditions in which hippocampal BDNF is deficient, lactoferrin supplementation could be a useful strategy to increase cellular BDNF production and improve neuronal development and circuitry.

We proposed that lactoferrin would mediate neuroprotective benefits in the FGR fetal sheep brain principally through its anti-inflammatory effects [25,26]. The microglia are the resident immune cells of the brain that respond to perinatal stressors in a context-dependent manner, commonly characterised by changes in their phenotypic characteristics [55]. Preclinical small- and large-animal studies have shown that placental insufficiency and FGR induce a neuroinflammatory response, with an increase in total microglial cell population and a shift in microglial phenotype towards an ameboid phenotype, both strongly attributed to perinatal neuropathology [11,49,55,56,57,58]. In this study, FGR did not produce an increase in microglia cell count within the hippocampal regions examined, nor did it increase the number of microglia with an ameboid phenotype (Figure 5). Previous studies by our group have shown that microglial cell number is increased in the FGR fetal brain at the same gestational age as assessed here, with the most pronounced effects seen in white matter regions, while the hippocampus was not examined [49]. This may indicate a greater susceptibility of cerebral white matter to neuroinflammation, which fits with the strong association between FGR and white matter injury in infants born preterm [12,59]. Despite an absence of neuroinflammation in the hippocampus in the FGR group, results show that antenatal lactoferrin treatment moderated microglial cell number and produced a shift in microglial phenotypes, reducing microglial cell count (Iba1+) in the CA3 region of AGA and FGR fetuses compared to untreated groups, and reducing the number of resting microglia in the CA3 region. We did not see the same effect in the CA1 region of the hippocampus. A recent study in adult mice challenged with lipopolysaccharide found that lactoferrin regulates hippocampal microglia by inhibiting the activation of the cell surface Toll-like receptor 4-nuclear factor κB complex [60]. In turn, an overall reduction in microglial cells may act to elicit the further effect of reduced brain pro-inflammatory cytokine release [32]. Interestingly, in the current study, we found that the total decrease in microglia was predominantly underpinned by a significant reduction in the resting microglial cell phenotype with lactoferrin treatment. The resting microglia represent the non-activated quiescent form of microglia, with a reduction in resting microglia usually associated with acute injury [45,55]. In the current study, it is not possible to further ascribe the anti-inflammatory benefits of lactoferrin, given that we did not see a pro-inflammatory status within the FGR cohort. Nonetheless, these results indicate that further research is warranted to characterise the actions of lactoferrin on microglia cell regulation.

NeuN immunostaining has traditionally been used to count mature neurons, but it has also been identified as the pre-mRNA alternative splicing protein Rbfox3 [61]. The Rbfox3 protein is critical for gene regulation, signalling, and intracellular communication [62]. As such, this study employed the profile of NeuN/Rbfox3 immunostaining to categorise neuronal morphology in the fetal sheep hippocampus. We have previously observed that FGR in newborn lambs was associated with no change in the total number of NeuN-positive hippocampal neurons, but with an increased proportion of unhealthy, chromatin-condensed neurons [63]. In the current study, at an earlier fetal gestational age, we did not find this to be the case in the FGR group. The effect of antenatal lactoferrin exposure on hippocampal neurons was interesting. We found a significant increase in the total number of neurons within both the CA1 and the CA3 regions of lactoferrin-treated AGA and FGR fetuses. Given that we showed increased BDNF protein in the CA1 and CA3 regions, we could speculate that increased BDNF mediates increased neuronal proliferation and survival. Wang et al. also demonstrated that lactoferrin promotes neuronal proliferation via upregulation of ERK1 and 2 phosphorylation [60]. These observations potentially support a neuroprotective effect of lactoferrin, although the increased number of cells with abnormal (clumped or aggregated) nuclear Rbfox3 suggests that these cells are not developmentally normal [44]. Intriguingly, the increase in NeuN-positive neurons in both the CA1 and CA3 regions following lactoferrin treatment was accompanied by a reduction in MAP-2 immunoreactivity across these regions, indicating reduced dendrite area, integrity, or cytoskeletal stability. The synergistic increase in NeuN-positive cell body density and decreased MAP2 following lactoferrin are strong indicators that the neuronal cell bodies are more tightly packed, facilitated by decreased dendrite outgrowth. The NeuN/RBfox3 protein stain allowed us to characterise neuronal morphology, and we found that lactoferrin-treated animals showed an increase in the proportion of abnormal chromatin neurons in the CA1 and CA3 regions, which displayed nuclear clumping or aggregation. Together, these results indicate that antenatal lactoferrin exposure increases hippocampal neuron proliferation or survival, but neurons are unlikely to be normal, showing an abnormal NeuN/RBfox3 immunoreactivity in the nucleus that is associated with degeneration [44], and with significantly reduced dendritic arbors.

At 127 days of gestation, the FGR fetuses had a reduced body weight, together with brain sparing, but we observed very limited adverse outcomes on the hippocampus at this timepoint. The lack of neuropathology observed in the hippocampus of the FGR cohort is most likely due to an insufficient duration of placental dysfunction and FGR, with our previous work showing a progressive worsening of brain injury over time in late pregnancy [8]. Lactoferrin did not induce growth-promoting effects on the fetus in either AGA or FGR cohorts. Previous studies of FGR in rat pups showed that maternal lactoferrin did not improve birth weight in growth-restricted offspring [32,53], but did induce postnatal catch-up growth by day 21 after birth [53]. The fetal study presented here administered lactoferrin for 32 days in late pregnancy, and in the newborn cohort, lactoferrin treatment was extended to 41 days in total. In the newborn cohort we found that FGR + Lacto lambs were more likely than FGR untreated lambs to require oxygen after birth and to do more poorly (slow to stand, very lethargic); we think it is likely that this is due to a raised haematocrit in lactoferrin-treated lambs. The number of animals included in the lamb cohort is low and we do not have the statistical power to detect group differences. Still, the observation of increased circulating haematocrit has been shown previously in a human trial of lactoferrin supplementation in neonates [64]. In the setting of FGR, where an increasing degree of fetal hypoxia is positively correlated with fetal red blood cell level [65], it would not be our recommendation to recommend a therapy that has the potential to increase haematocrit further and risk the induction of polycythemia.

A primary limitation of the experimental design utilised here is that FGR did not induce measurable neuropathology within the CA1 and CA3 regions of the hippocampus. This is surprising, as in previous studies we have shown significant brain pathology, oxidative stress, and inflammation at a similar late-preterm fetal sheep gestation, although this was predominantly confined to the white matter brain regions [11,49]. The absence of hippocampal injury in this cohort may reflect the heterogeneous nature of FGR, including variability in the severity, timing, and etiology of placental insufficiency, as well as species-specific resilience and the timing of assessment. Lactoferrin has also been shown to protect or normalise white matter development in the perinatal brain [32], and therefore, assessment of the white matter in the current FGR and AGA cohort is a next step. A lack of pathology in the hippocampus of FGR fetuses did not allow for the assessment of the neuroprotective actions of antenatal lactoferrin. The dose of lactoferrin administered to the ewes during pregnancy was derived from small-animal experiments [31] and adapted for large animals. The optimum neuroprotective dose of lactoferrin administered during pregnancy or postnatally is not currently known. We administered lactoferrin orally; the digestive system of a sheep is different from that of humans (and rodents as per previous studies), given that sheep are ruminants, and therefore, we do not know if this dose of 36 g per day was optimal. Certainly, in this study, it is evident from effects on the brain that lactoferrin crossed the placenta and the fetal blood–brain barrier. The effects of lactoferrin on the immature brain are dose-dependent, with a dosing study in hypoxic–ischaemic neonatal rats showing that excessive concentrations of lactoferrin may be harmful for cerebral development [31], and we cannot rule out that the dose used in the current study was too high. The tissue for the present study was collected and processed for histological and immunofluorescence analyses, precluding additional molecular or quantitative assays such as RT-qPCR, Western blotting, or functional microglial assessments. Consequently, the outcomes of the current study are based, in part, on semi-quantitative measures. Tissue was also collected for Golgi analysis of neuronal complexity, reported separately. While quantitative molecular validation could strengthen the findings, the immunohistochemical measures used here, together with the observed physiological outcomes, provide meaningful insights into the effects of maternal lactoferrin on the fetal brain. A further potential limitation of this study is that the use of twins resulted in an unequal distribution of lactoferrin between the fetuses. Because SUAL restricts blood flow to the placenta of the growth-restricted twin, transplacental transfer of lactoferrin may have been reduced compared with the AGA twin. This differential exposure introduces the possibility that the observed outcomes reflect not only biological differences in treatment response but also disparities in drug delivery.

## 5. Conclusions

Antenatal lactoferrin supplementation produced significant effects within the fetal brains of both AGA and FGR offspring, which are consistent with neuroprotective benefits. Most notably, lactoferrin increased hippocampal BDNF expression, increased NeuN-positive neuronal cell count, and reduced the total number of microglial cells. Interestingly, despite these positive effects on neuronal survival, MAP-2 expression in hippocampal neurons was reduced, and the number of abnormal chromatin neurons (Rbfox3-clumping) increased, indicating that neuronal integrity was diminished. We propose that lactoferrin exhibits modulatory actions on the developing brain that are potentially beneficial under conditions of acute hypoxic or inflammatory stress in the perinatal period. Further evidence is required to define whether an optimal dosing window can maximise lactoferrin’s neuroprotective potential while minimising the risk of adverse side effects. However, overall, our results indicate that antenatal lactoferrin is not the neuroprotective agent of choice for prolonged antenatal treatment required for FGR, given its effects of exacerbating polycythemia.

## Figures and Tables

**Figure 1 cells-14-01951-f001:**
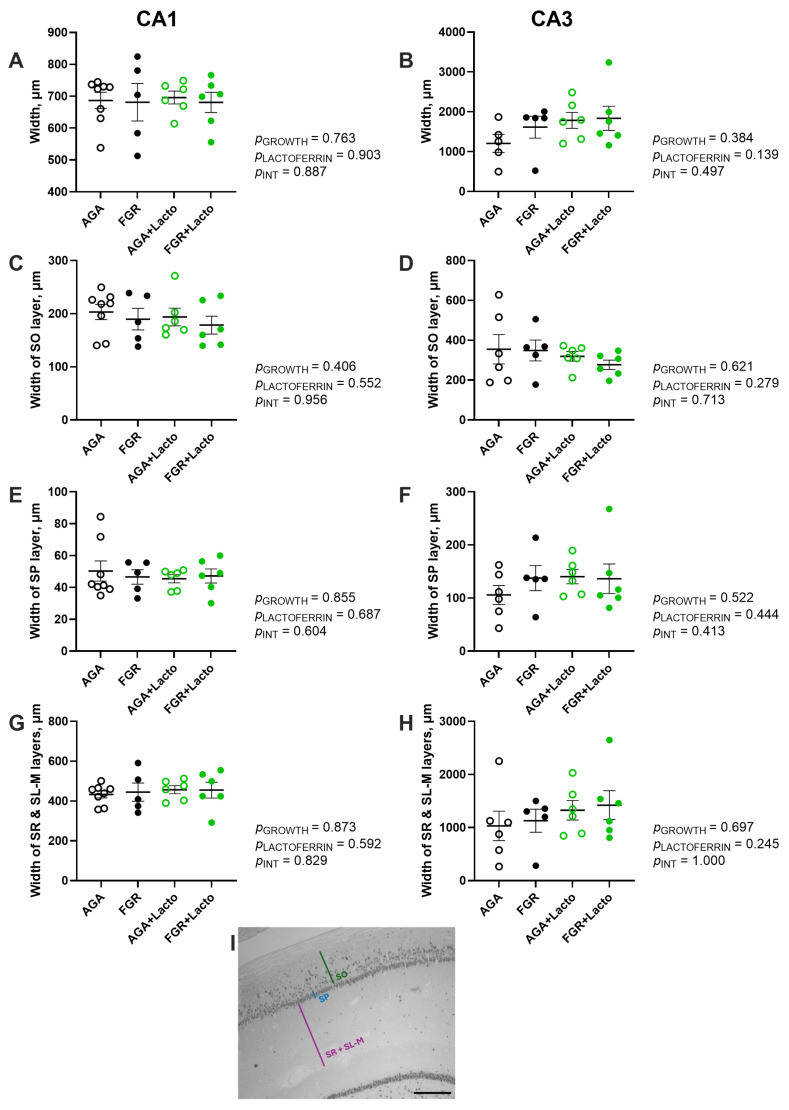
Hippocampal layer width. The widths of the total (**A**,**B**), stratum oriens (SO; (**C**,**D**)), stratum pyramidal (SP; (**E**,**F**)), and combined stratum radiatum (SR) and stratum lacunosum moleculare (SL-M; (**G**,**H**)) layers were assessed in the CA1 and CA3 regions. Identification of individual strata layers (**I**). Groups are AGA (black non-filled, n = 8), FGR (black, n = 5), AGA + Lacto (green non-filled, n = 6), and FGR + Lacto (green filled, n = 6); individual data points are presented, and mean ± SEM values. Two-way ANOVA, with Tukey’s post hoc (*p* < 0.05), was used for statistical analyses. Scale bar = 250 µm.

**Table 1 cells-14-01951-t001:** Fetal weights.

	AGA	FGR	AGA + Lacto	FGR + Lacto	
n (M/F)	8 (5/3)	5 (3/2)	6 (3/3)	6 (2/4)	
Body weight, Kg	3.74 ± 0.21	2.95 ± 0.24	3.43 ± 0.26	2.36 ± 0.32	***p*_GROWTH_ = 0.002** *p*_LACTO_ = 0.101 *p*_INT_ = 0.607
Brain weight, g	46.57 ± 0.76	46.10 ± 1.96	47.54 ± 2.31	41.55 ± 2.64	*p*_GROWTH_ = 0.110 *p*_LACTO_ = 0.365 *p*_INT_ = 0.168
Brain–body weight, g/Kg	12.68 ± 0.65	15.91 ± 1.01	14.15 ± 0.94	18.78 ± 1.80	***p*_GROWTH_ = 0.003** *p*_LACTO_ = 0.072 *p*_INT_ = 0.546
Brain–liver weight, g/g	0.50 ± 0.06	0.62 ± 0.06	0.50 ± 0.06	0.73 ± 0.12	***p*_GROWTH_ = 0.032** *p*_LACTO_ = 0.471 *p*_INT_ = 0.519

Data presented as mean ± SEM and analysed using two-way ANOVA, with Tukey’s post hoc (*p* < 0.05). Bold indicates significant differences.

**Table 2 cells-14-01951-t002:** Newborn lamb weights.

	24 h			4 Weeks		
	AGA	FGR	FGR + Lacto	AGA	FGR	FGR + Lacto
n (M/F)	8 (3/5)	9 (4/5)	5 (2/3)	12 (8/4)	9 (5/4)	4 (0/4)
Body weight, Kg	5.03 ± 0.34	3.25 ± 0.35 **	3.22 ± 0.10 **	13.40 ± 0.38	11.06 ± 0.50 *	10.15 ± 1.37 **
Brain–body weight, g/Kg	11.02 ± 0.86	16.52 ± 1.72 *	15.20 ± 0.44	5.56 ± 0.16	6.51 ± 0.26	7.20 ± 1.11 *
Brain–liver weight, g/g	0.380 ± 0.040	0.582 ± 0.061 *	0.531 ± 0.037	0.258 ± 0.008	0.297 ± 0.019	0.348 ± 0.063
Heart–body weight, g/Kg	8.81 ± 0.40	8.76 ± 0.29	9.81 ± 0.83	6.91 ± 0.35	7.19 ± 0.45	6.24 ± 0.42
Lung–body weight, g/Kg	26.35 ± 1.50	23.63 ± 1.65	23.87 ± 2.90	20.92 ± 2.01	19.85 ± 2.05	17.93 ± 1.86

N.B. Newborn data is limited due to small group sizes and early fatalities in FGR + Lacto group. Data presented as mean ± SEM and analysed using one-way ANOVA with Tukey’s multiple comparison. * *p* < 0.05, ** *p* < 0.01 vs. AGA within age group.

## Data Availability

The data presented in this study are available on request from the corresponding author.

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
