# Peer review of "Does Antenatal Lactoferrin Protect Hippocampal Development in Ovine Fetuses with Growth Restriction?"

_cells, 2025, doi:10.3390/cells14241951_

Round 1
Reviewer 1 Report
Comments and Suggestions for Authors
This manuscript outlines a meticulously designed and well-conducted preclinical investigation of the neuroprotective effects of maternal lactoferrin supplementation in fetal sheep with induced fetal growth restriction (FGR). The experimental design, which incorporates various control and treatment groups, appropriate timing for the intervention, and comprehensive histological evaluations, serves as a strong model for translational research. The paper is well written and offers substantial methodological detail, along with a balanced interpretation of the results. The study provides valuable insights into the potential benefits and limitations of antenatal lactoferrin administration for fetal brain development. However, a few modifications and clarifications are required prior to publication. Thorough experimental design utilizing a validated ovine model of early-onset fetal growth restriction (FGR). - Extensive immunohistochemical and morphometric analysis of hippocampal subregions (CA1 and CA3). - A well-rounded discussion that considers both neuroprotective effects (↑BDNF, ↓microglia) and potential adverse outcomes (↑abnormal chromatin, ↓MAP2). - Clear explanation of ethical considerations and statistical rigor. - Well-supported translational relevance.
Minor Change Remarks Abstract - Revise the conclusion for neutrality: "Prolonged antenatal lactoferrin exposure may yield mixed effects on fetal brain development and hematological balance." - Include the sample size and clarify that no significant hippocampal damage was observed in the FGR animals.
Introduction: Include a brief explanation of why sheep serve as an effective model for studying the prenatal development of the human brain, highlighting similarities such as the concurrent maturation of the hippocampus and placental physiology. Explicitly identify the knowledge gap by stating that the long-term effects of lactoferrin on the developing hippocampus have not been previously evaluated in large animal models.
Methods: Please provide the body weights or weight ranges of the ewes used to justify the lactoferrin dosage of 36 g/day (approximately 0.5 g/kg/day). Additionally, clarify whether the fetal sex ratio was balanced across all groups, as this may influence neurodevelopmental outcomes. The description of the pilot study confirming the placental transfer of lactoferrin should be either relocated or succinctly summarized, with specific details placed in the Supplementary Material. It must also be noted that all histological assessments were conducted without knowledge of the treatment group assignments.
. Results: For Figures 2–4, ensure that all scale bars and labels (CA1, CA3) are present and accurate. In Table 2, clarify that the newborn data are limited due to small group sizes and early fatalities in the FGR+Lacto group. Regarding the neuronal morphology findings, specify whether the increased neuronal abundance includes both morphologically abnormal neurons and solely NeuN-positive counts.
Discussion: Expand the discourse on the mechanisms underlying the reduction of MAP2 in the context of elevated neuronal counts. Consider referencing the potential overactivation of BDNF/ERK or the cytoskeletal instability associated with stress. Additionally, further elaborate on the lack of identified hippocampal pathology in FGR fetuses. This absence may be attributable to the duration of placental insufficiency, species-specific resilience, or the timing of the assessment. Clearly articulate the implications for humans, focusing on how to determine the appropriate dosage, the optimal timing for administration, and the safety profile for pregnant women. It may also be beneficial to include a brief comparison to studies involving postnatal lactoferrin supplementation in other animal models, such as rodents and piglets.
Tables and Figures: Ensure that all figure legends contain comprehensive details regarding the statistical tests employed, including sample sizes (n values), the type of ANOVA performed, and any post hoc tests applied. Furthermore, enhance the resolution and contrast of histological images. In the figure captions, indicate which results were statistically significant.
Make minor adjustments to improve the tone and clarity of the text. For example, replace "Lactoferrin mediated" with "Lactoferrin exerted," and revise the sentence "After that, standard paraffin embedding was done" to "Samples were then paraffin-embedded using standard protocols." Ensure that all terms, such as SUAL and MAP2, are defined upon their first use and that all abbreviations are consistently used.
This study provides valuable insights into the effects of lactoferrin on fetal hippocampal development during pregnancy. This paper will significantly contribute to the fields of perinatal neuroprotection and placental insufficiency research. With some minor textual revisions, an expanded discussion of the underlying mechanisms, and improved figures, the work will be even more impactful.
Author Response
Reviewer 1
This manuscript outlines a meticulously designed and well-conducted preclinical investigation of the neuroprotective effects of maternal lactoferrin supplementation in fetal sheep with induced fetal growth restriction (FGR). The experimental design, which incorporates various control and treatment groups, appropriate timing for the intervention, and comprehensive histological evaluations, serves as a strong model for translational research. The paper is well written and offers substantial methodological detail, along with a balanced interpretation of the results. The study provides valuable insights into the potential benefits and limitations of antenatal lactoferrin administration for fetal brain development. Thank you for the thoughtful and balanced evaluation. We appreciate the recognition of the study design, methodological detail, and translational intent. The point about both the promise and limits of lactoferrin as a therapy is correct. We hope the revisions to this manuscript ensure the conclusions remain measured and well supported.
However, a few modifications and clarifications are required prior to publication. Thorough experimental design utilising a validated ovine model of early-onset fetal growth restriction (FGR). - Extensive immunohistochemical and morphometric analysis of hippocampal subregions (CA1 and CA3). - A well-rounded discussion that considers both neuroprotective effects (↑BDNF, ↓microglia) and potential adverse outcomes (↑abnormal chromatin, ↓MAP2). - Clear explanation of ethical considerations and statistical rigor. - Well-supported translational relevance.
- Minor Change Remarks Abstract - Revise the conclusion for neutrality: "Prolonged antenatal lactoferrin exposure may yield mixed effects on fetal brain development and hematological balance." - Include the sample size and clarify that no significant hippocampal damage was observed in the FGR animals.
The concluding remarks have been modified. Both sample size (line 23-4) and the lack of hippocampal difference (line 25) are noted in the abstract
- Introduction: Include a brief explanation of why sheep serve as an effective model for studying the prenatal development of the human brain, highlighting similarities such as the concurrent maturation of the hippocampus and placental physiology. Explicitly identify the knowledge gap by stating that the long-term effects of lactoferrin on the developing hippocampus have not been previously evaluated in large animal models.
The gestation length and profile of brain development are long in sheep, compared to small animals that may otherwise be the subject of FGR studies, which is particularly useful when aiming to characterise prolonged brain developmental processes, such as dendritogenesis in the hippocampus. It is important to note that, at normal term birth, the sheep brain is more mature than in human infants, and at least in part this was offset in the current set of studies by inducing moderate preterm birth in all lambs.
As the hippocampus was the brain region of focus of this manuscript, the protracted development of this brain region needed to be adequately considered. The hippocampus has a complex and relatively prolonged developmental profile, beginning early in gestation and continuing into adulthood, with a peak developmental growth spurt for substantial dendritogenesis occurring in the final trimester of human pregnancy and into neonatal life. The benefit of utilising sheep is their longitudinal brain development, a pattern that mimics human hippocampal development.
We have added the following to the introduction (line 65-75):
“Sheep offer a practical and biologically relevant model for prenatal brain development because their gestation length and trajectory of neural maturation are far closer to humans than those of small-animal models. Their prolonged, longitudinal pattern of brain growth mirrors key human processes, including the extended maturation of the hippocampus and the parallel demands placed on placental physiology during late gestation. Although the sheep brain is slightly more mature at term than the human newborn brain, the moderately preterm stage used in this study is well matched to late-preterm human brain development. Given the hippocampus was our primary region of interest, and its dendritic growth peaks late in gestation and extends into early life, the developmental timing in sheep provides an advantageous window for examining these protracted neural processes.”
And we have expanded line 85 to explicitly state the novelty of using a large animal model.
- Methods: Please provide the body weights or weight ranges of the ewes used to justify the lactoferrin dosage of 36 g/day (approximately 0.5 g/kg/day). Additionally, clarify whether the fetal sex ratio was balanced across all groups, as this may influence neurodevelopmental outcomes. The description of the pilot study confirming the placental transfer of lactoferrin should be either relocated or succinctly summarized, with specific details placed in the Supplementary Material. It must also be noted that all histological assessments were conducted without knowledge of the treatment group assignments.
We have added the mean maternal weight (line 145). The ratio of males vs females is stated in Table 1 and not statistically different across groups. We have removed the reference to the pilot study.
- Results: For Figures 2–4, ensure that all scale bars and labels (CA1, CA3) are present and accurate. In Table 2, clarify that the newborn data are limited due to small group sizes and early fatalities in the FGR+Lacto group. Regarding the neuronal morphology findings, specify whether the increased neuronal abundance includes both morphologically abnormal neurons and solely NeuN-positive counts.
We have double checked the figures and all scale bars and labels are correct, added the suggested wording to Table 2 and added the word ‘total’ to line 250 to ensure clarity on neuronal findings.
- Discussion:
- Expand the discourse on the mechanisms underlying the reduction of MAP2 in the context of elevated neuronal counts. Consider referencing the potential overactivation of BDNF/ERK or the cytoskeletal instability associated with stress.
Thank you for noting this interesting point. As you alluded to, MAP2 is an abundant neuronal cytoskeletal protein that provides structural stability, and its immunohistochemical staining is used to quantify neuronal cell bodies and dendrites. Lactoferrin treatment, per se, induced an interesting pattern of change in both the CA1 and CA3 fields, increasing NeuN-positive cell bodies, decreasing MAP2, and increasing BDNF expression. The increase in NeuN and decrease in MAP2 sit logically together, with an apparent increase in the packing density of neuronal cell bodies as the dendritic cytoskeletons are contracted. The actions of the BDNF protein are modulated via its high-affinity receptor TrkB, and others have shown differential regulation of BDNF and TrkB in response to FGR. Unfortunately, we were unable to obtain an ovine-specific TrkB antibody for this study, but it remains an important follow-up.
We have added two lines to elaborate these points:
(Line 384-386): ‘BDNF and TrkB expression may be differentially expressed in FGR (52), and we were not able to examine TrkB in the current study, which we acknowledge is a limitation for interpretation of BDNF.’
(Line 452-454): ‘The synergistic increase in NeuN-positive cell body density and decreased MAP2 following lactoferrin are strong indicators that the neuronal cell bodies were more tightly packed, facilitated by decreased dendrite outgrowth’.
- Additionally, further elaborate on the lack of identified hippocampal pathology in FGR fetuses. This absence may be attributable to the duration of placental insufficiency, species-specific resilience, or the timing of the assessment. We thank the reviewer for this comment. The lack of identified hippocampal pathology in FGR fetuses is now addressed more thoroughly in the discussion. We have elaborated on potential reasons for this finding, including the relatively short duration of placental insufficiency in the current model, species-specific resilience of the ovine hippocampus, and the timing of tissue collection at late-preterm gestation. We also reference differences in experimental design compared with previous studies, which may account for discrepancies in observed neuropathology.
Added (line 371): “The absence of hippocampal pathology in the current study may reflect the relatively short duration of placental insufficiency, species-specific resilience of the ovine hippocampus, or the timing of tissue collection at late-preterm gestation. Additionally, differences in experimental design parameters, such as the severity and timing of FGR induction, may account for these discrepancies.”
Line 486: “The absence of hippocampal injury in this cohort may reflect the heterogeneous nature of FGR, including variability in the severity, timing, and etiology of placental insufficiency, as well as species-specific resilience and the timing of assessment.”
- Clearly articulate the implications for humans, focusing on how to determine the appropriate dosage, the optimal timing for administration, and the safety profile for pregnant women. We thank the reviewer for their comments regarding translation to humans. Based on our findings, chronic antenatal lactoferrin is not a viable therapeutic option for prolonged treatment in FGR, as it exacerbates polycythemia. Therefore, no further studies on dosage optimisation or clinical translation are planned, and we believe such considerations are beyond the scope of this manuscript. We have amended the penultimate sentence in the conclusion to reflect this.
“Further evidence is required to define whether an optimal dosing window can maximise lactoferrin’s neuroprotective potential while minimising the risk of adverse side effects. However, overall, our results indicate that antenatal lactoferrin is not the neuroprotective agent of choice for prolonged antenatal treatment required for FGR, given its effects to exacerbate polycythemia.”
- It may also be beneficial to include a brief comparison to studies involving postnatal lactoferrin supplementation in other animal models, such as rodents and piglets. We thank the reviewer for the suggestion. However, the focus of the current study is antenatal lactoferrin supplementation. Including a discussion of postnatal administration in other species would be outside the scope of this manuscript and could confuse the central message. Therefore, we have not added this comparison.
- Tables and Figures: Ensure that all figure legends contain comprehensive details regarding the statistical tests employed, including sample sizes (n values), the type of ANOVA performed, and any post hoc tests applied. Furthermore, enhance the resolution and contrast of histological images. In the figure captions, indicate which results were statistically significant. We have addressed this request with the caveat that the statistical outcomes are already directly represented on the graphs and described in the Results section, and adding this information to the figure captions would make them unnecessarily long and repetitive. The existing presentation clearly indicates all significant findings, so we have retained the current caption format.
- Make minor adjustments to improve the tone and clarity of the text. For example, replace "Lactoferrin mediated" with "Lactoferrin exerted," and revise the sentence "After that, standard paraffin embedding was done" to "Samples were then paraffin-embedded using standard protocols." Ensure that all terms, such as SUAL and MAP2, are defined upon their first use and that all abbreviations are consistently used.
Thank you, these have been amended.
- This study provides valuable insights into the effects of lactoferrin on fetal hippocampal development during pregnancy. This paper will significantly contribute to the fields of perinatal neuroprotection and placental insufficiency research. With some minor textual revisions, an expanded discussion of the underlying mechanisms, and improved figures, the work will be even more impactful. Thank you.
Reviewer 2 Report
Comments and Suggestions for Authors
Dear Editors,
I'm pleased to have the opportunity to review article entitled: Does antenatal lactoferrin protect hippocampal development in ovine fetuses with growth restriction?
The manuscript presents a well-designed and carefully executed experimental study exploring the neuroprotective potential of antenatal lactoferrin supplementation in a large-animal model of early-onset fetal growth restriction (FGR). The authors address a clinically relevant question, as FGR remains a major cause of perinatal morbidity and long-term neurodevelopmental impairment, with limited preventive or therapeutic options available. The study’s novelty lies in the antenatal administration of lactoferrin to the ewe and the subsequent assessment of hippocampal structure, neuronal integrity, neuroinflammatory markers, and oxidative stress parameters in the fetal and neonatal brain.
However there are couple of issues:
Introduction:
The final paragraph effectively introduces the study aims but could more clearly articulate the novelty—specifically that this is the first study to assess chronic maternal lactoferrin supplementation and hippocampal neurodevelopment in a large-animal model of FGR.
Additionally:
-
Ensure consistency in terminology (e.g., “appropriate for gestational age [AGA]” vs. “control”).
-
Clarify whether oxidative stress markers were quantitatively assessed or inferred indirectly.
-
A brief statement on maternal tolerance or systemic effects of lactoferrin supplementation would be valuable.
-
Ensure all p-values are presented to three decimal places where possible.
-
Consider rephrasing “hippocampal structure was unaffected” to “no significant differences in hippocampal layer width were detected,” which avoids implying absence of any structural change.
-
The reference list appears comprehensive; ensure all cited works (especially [13], [31], [39–41]) are current and correctly formatted.
References:
-
Check all author initials for consistency.
-
Remove page numbering in the text if it is not part of the citation.
-
Standardize DOI formatting (without a period at the end).
-
Harmonize journal abbreviation style and italics.
Author Response
Reviewer 2
The manuscript presents a well-designed and carefully executed experimental study exploring the neuroprotective potential of antenatal lactoferrin supplementation in a large-animal model of early-onset fetal growth restriction (FGR). The authors address a clinically relevant question, as FGR remains a major cause of perinatal morbidity and long-term neurodevelopmental impairment, with limited preventive or therapeutic options available. The study’s novelty lies in the antenatal administration of lactoferrin to the ewe and the subsequent assessment of hippocampal structure, neuronal integrity, neuroinflammatory markers, and oxidative stress parameters in the fetal and neonatal brain. Thank you for the positive feedback.
However there are couple of issues:
Introduction: The final paragraph effectively introduces the study aims but could more clearly articulate the novelty—specifically that this is the first study to assess chronic maternal lactoferrin supplementation and hippocampal neurodevelopment in a large-animal model of FGR. We have amended the final paragraph to reiterate the novelty.
Added (Line 90): “The antenatal neuroprotective efficacy of lactoferrin remains largely unknown, particularly when administered chronically, and has never been studied in a long-term large animal model.”
Additionally:
- Ensure consistency in terminology (e.g., “appropriate for gestational age [AGA]” vs. “control”). We have double checked and amended.
- Clarify whether oxidative stress markers were quantitatively assessed or inferred indirectly. Oxidative stress was directly quantified following immunohistochemical detection of 8-OHdG in tissue sections. Staining was quantified across experimental groups to allow comparison of oxidative stress levels. However, given other reviewers concerns we have now removed the oxidative stress outcomes from this manuscript.
- A brief statement on maternal tolerance or systemic effects of lactoferrin supplementation would be valuable. We have added the following to the methods (line 135) “All ewes readily consumed the lactoferrin-supplemented feed, and no adverse effects or changes in general health were observed.”
- Ensure all p-values are presented to three decimal places where possible. I have checked all p values are three decimal places, with the exception of standard p values 0.05, 0.01 and 0.001
- Consider rephrasing “hippocampal structure was unaffected” to “no significant differences in hippocampal layer width were detected,” which avoids implying absence of any structural change. Amended.
- The reference list appears comprehensive; ensure all cited works (especially [13], [31], [39–41]) are current and correctly formatted. Thank you, references have been checked and amended as below
References:
- Check all author initials for consistency.
- Remove page numbering in the text if it is not part of the citation.
- Standardize DOI formatting (without a period at the end).
- Harmonize journal abbreviation style and italics.
Reviewer 3 Report
Comments and Suggestions for Authors
This study is highly valuable as it investigates the effects of maternal lactoferrin administration during the fetal period using a sheep model of human fetal growth restriction (FGR) induced by single umbilical artery ligation. However, since the analysis relies solely on immunohistochemistry, it is difficult to comprehensively evaluate the biological effects of lactoferrin. Specific comments are provided below.
-
The scientific name of the experimental animal should be clearly stated. It is presumably Ovis aries, but this should be explicitly indicated.
-
Detailed descriptions of the experimental design and animal procedures in accordance with the ARRIVE guidelines are lacking.
-
The source of the lactoferrin used in this study should be specified—whether it was derived from bovine, ovine, or another species.
-
It should be clarified whether the control group (without lactoferrin treatment) also received 12.5% glycerin, consistent with the treatment group.
-
Since most of the experimental results are based on immunofluorescence staining, additional quantitative analyses—such as RT-qPCR using laser microdissected tissue samples or Western blotting—should be performed to strengthen the findings.
-
NeuN staining should be performed in conjunction with nuclear staining, such as Hoechst 33258, to confirm neuronal localization.
-
The conclusion that maternal lactoferrin administration increases fetal brain BDNF expression is not sufficiently supported by immunostaining data alone. Additional quantitative validation is required.
-
Co-localization images using double immunostaining for Iba1 and 8-OHdG should be included to provide further evidence.
-
Although 8-OHdG is a marker of oxidative stress and inflammation, it does not reflect functional aspects such as cytokine production or phagocytic activity. At least one or two microglia-specific markers, activation markers, or inflammatory indicators (at the mRNA or protein level) should be analyzed to provide a more comprehensive assessment.
-
Because BDNF is a secreted neurotrophic factor that is rapidly released into the extracellular space, its intracellular accumulation is limited, and immunohistochemical detection tends to yield weak signals. Therefore, evaluating BDNF expression solely by IHC is not appropriate; quantitative assays are recommended.
Author Response
Reviewer 3
Comments and Suggestions for Authors
This study is highly valuable as it investigates the effects of maternal lactoferrin administration during the fetal period using a sheep model of human fetal growth restriction (FGR) induced by single umbilical artery ligation. However, since the analysis relies solely on immunohistochemistry, it is difficult to comprehensively evaluate the biological effects of lactoferrin. Specific comments are provided below.
- The scientific name of the experimental animal should be clearly stated. It is presumably Ovis aries, but this should be explicitly indicated. Added, thank you.
- Detailed descriptions of the experimental design and animal procedures in accordance with the ARRIVE guidelines are lacking. We thank the reviewer for highlighting this. Detailed descriptions of the experimental design and animal procedures have been provided in accordance with the ARRIVE guidelines, and the completed ARRIVE checklist has been submitted with the manuscript.
- The source of the lactoferrin used in this study should be specified—whether it was derived from bovine, ovine, or another species. We thank the reviewer for this comment. While the source of lactoferrin is mentioned in the paragraph title, we will explicitly state in the text that the lactoferrin used in this study was bovine-derived to avoid any ambiguity (Line 131).
- It should be clarified whether the control group (without lactoferrin treatment) also received 12.5% glycerin, consistent with the treatment group. A subset of control animals received 12.5% glycerin, and no differences in key outcomes were observed compared to saline-treated controls. To align with the ARRIVE guidelines and the principles of Replacement, Reduction, and Refinement (3Rs), we subsequently used saline-treated animals as controls to allow ewes to be used across multiple studies.
- NeuN staining should be performed in conjunction with nuclear staining, such as Hoechst 33258, to confirm neuronal localization. NeuN immunofluorescence provides a specific nuclear marker for neurons, and we performed detailed characterisation of neuronal phenotypes by strictly following methodology published in this journal (https://doi.org/10.3390/cells12202454). Adding a nuclear counterstain such as Hoechst would interfere with this phenotypic analysis, which relies on subtle NeuN signal patterns to distinguish healthy, degenerating, and abnormal neurons.
- Co-localization images using double immunostaining for Iba1 and 8-OHdG should be included to provide further evidence. For ease of interpretation of this data, we have removed the 8-OHdG data from this study.
- Since most of the experimental results are based on immunofluorescence staining, additional quantitative analyses—such as RT-qPCR using laser microdissected tissue samples or Western blotting—should be performed to strengthen the findings.
- The conclusion that maternal lactoferrin administration increases fetal brain BDNF expression is not sufficiently supported by immunostaining data alone. Additional quantitative validation is required.
- Although 8-OHdG is a marker of oxidative stress and inflammation, it does not reflect functional aspects such as cytokine production or phagocytic activity. At least one or two microglia-specific markers, activation markers, or inflammatory indicators (at the mRNA or protein level) should be analyzed to provide a more comprehensive assessment.
- Because BDNF is a secreted neurotrophic factor that is rapidly released into the extracellular space, its intracellular accumulation is limited, and immunohistochemical detection tends to yield weak signals. Therefore, evaluating BDNF expression solely by IHC is not appropriate; quantitative assays are recommended.
We thank the reviewer for these insightful comments. We acknowledge that additional quantitative analyses, such as RT-qPCR using laser microdissected tissue samples, Western blotting, or microglia-specific functional assays, could provide complementary validation of the immunofluorescence findings. However, the tissue collected in this study was processed explicitly for histological and immunofluorescence analyses and is not suitable for molecular or protein-based assays. Tissue was also collected for Golgi analysis of neuronal complexity, which is reported in a separate manuscript. Consequently, additional molecular or quantitative analyses were not feasible for the present cohort.
In response to the reviewer's comments on oxidative stress, we note that our 8-OHdG staining showed no differences between groups, and this measure has therefore been removed from the manuscript. While we acknowledge the inherent limitations of BDNF detection via immunohistochemistry, previous studies in sheep have demonstrated that IHC alone can reliably detect changes in BDNF expression (PMID: 19050322, PMID: 39737892), supporting the validity of our approach. Importantly, the observed patterns, particularly evidence of poor postnatal adaptation, remain biologically meaningful.
We will include the following in the limitations (line 491-498): “The tissue for the present study was collected and processed for histological and immunofluorescence analyses, precluding additional molecular or quantitative assays such as RT-qPCR, Western blotting, or functional microglial assessments. Tissue was also collected for Golgi analysis of neuronal complexity, reported separately. While quantitative molecular validation could strengthen the findings, the immunohistochemical measures used here, together with observed physiological outcomes, provide meaningful and ethically justifiable insights into the effects of maternal lactoferrin on the fetal brain.”
Round 2
Reviewer 2 Report
Comments and Suggestions for Authors
Accept in present form
Author Response
Thank you for taking the time to review.
Reviewer 3 Report
Comments and Suggestions for Authors
Thank you very much for your revised manuscript. We appreciate the substantial improvements you have made to the content, and it is clear that considerable effort has been devoted to strengthening the work.
However, despite these improvements, the revision did not include sufficient, point-by-point responses to the reviewers’ comments. Clear and complete replies are essential for us to understand how each concern raised during the review process has been addressed. In the absence of such responses, it is difficult to properly evaluate the adequacy of the revisions and to ensure that all issues have been fully resolved.
Because of this lack of detailed responses to the review comments, we are unable to proceed with acceptance at this stage. We regret to inform you that we must therefore decline the manuscript.
Supplement:
The authors' responses do not directly address the reviewers' requests for additional quantitative verification and supporting data. Consequently, they fail to provide sufficient evidence to support the conclusions in the main text. In particular, the assertion that “low-dose glycerol administration has absolutely no effect on sensitive organs such as the developing hippocampus” is insufficient. Since glycerol can cross the maternal-fetal barrier, direct biological indicators for both the mother and fetus (e.g., blood concentrations, behavioral assessments, neuropathological evaluations) must be presented. The authors argue for reusing animals citing “reuse” and the 3Rs as reasons. However, if no harm from glycerin is recognized, using the same individuals for additional analyses is reasonable. Simply dismissing this as a matter of “ethical consideration” is insufficient.
I request the author to provide clear and specific supplementary information and implementation regarding the following points. If additional experiments are physically impossible, I require a detailed explanation of this fact and appropriate revision of the conclusions and discussion in the main text of the paper.
1. The authors only state that “tissues were processed for IHC, making molecular analysis impossible,” which does not address the reviewers' concern (lack of quantifiability).
2. If possible, confirm whether the same cohort or other stored samples (e.g., frozen tissue, plasma) exist. This would allow for quantitative analysis of BDNF (via WB or ELISA) and mRNA/protein quantification of microglia/microglial activation markers (e.g., Iba1, CD68, P2RY12, TMEM119).
3. Even when drawing conclusions based solely on IHC, it is necessary to demonstrate semi-quantitative analysis (signal intensity measurement per ROI, cell counting, standardization procedures between slides, antibody validation data = blocking peptide and negative controls).
4. The authors counter that “previous studies in sheep demonstrate detectability using IHC alone,” but the cited literature does not guarantee the validity of IHC in this study. It is necessary to explicitly state whether experimental conditions such as antibodies, fixation conditions, and developmental stages are consistent.
Author Response
The authors' responses do not directly address the reviewers' requests for additional quantitative verification and supporting data. Consequently, they fail to provide sufficient evidence to support the conclusions in the main text. In particular, the assertion that “low-dose glycerol administration has absolutely no effect on sensitive organs such as the developing hippocampus” is insufficient. Since glycerol can cross the maternal-fetal barrier, direct biological indicators for both the mother and fetus (e.g., blood concentrations, behavioral assessments, neuropathological evaluations) must be presented. The authors argue for reusing animals citing “reuse” and the 3Rs as reasons. However, if no harm from glycerin is recognized, using the same individuals for additional analyses is reasonable. Simply dismissing this as a matter of “ethical consideration” is insufficient. We thank the reviewer for raising the issue of vehicle effects and agree that appropriate controls are essential when interpreting any neurodevelopmental outcome. To address this, we have provided data from an additional cohort in which offspring were exposed to maternal low-dose glycerine vehicle alone (see Fig. 1). In this cohort, low-dose glycerine treatment had no detectable impact on fetal hippocampal width, NeuN abundance, or Iba1-positive cell number. These data directly support our statement that low-dose glycerine does not alter key structural or cellular markers within the developing hippocampus.
We acknowledge the reviewer’s point that glycerol can cross the maternal–fetal barrier. The expanded data now provide direct evidence that, at the dose and timing used in this study, there is no measurable impact on hippocampal integrity. We also appreciate that broader physiological or behavioural endpoints could be considered; however, the parameters assessed here reflect the primary neuropathological outcomes relevant to our study design.
Regarding the use of vehicle-exposed animals, we recognise the importance of transparency around experimental limitations. We have also now included an explicit statement in the Methods noting the limitation of not having vehicle controls within the original cohort.
Added Line 136 “Control ewes received water only. In an earlier pilot study, a cohort of ewes were treated with the glycerine vehicle only, and we detected no differences in hippocampal width, NeuN abundance, or Iba1+ve cell abundance (data not shown); we therefore did not pursue a full group of vehicle animals and while this is a study limitation, it is unlikely to influence the study outcomes.”
Fig 1. Hippocampal width, neuronal abundance (NeuN+) and Iba+ cell abundance in the CA3 region of the hippocampus of control fetal sheep treated with saline or glycerine.
I request the author to provide clear and specific supplementary information and implementation regarding the following points. If additional experiments are physically impossible, I require a detailed explanation of this fact and appropriate revision of the conclusions and discussion in the main text of the paper.
- The authors only state that “tissues were processed for IHC, making molecular analysis impossible,” which does not address the reviewers' concern (lack of quantifiability). We believe that altering the wording throughout the document to indicate that the IHC analysis allows semi-quantitative assessment (see query 3) addresses this comment.
- If possible, confirm whether the same cohort or other stored samples (e.g., frozen tissue, plasma) exist. This would allow for quantitative analysis of BDNF (via WB or ELISA) and mRNA/protein quantification of microglia/microglial activation markers (e.g., Iba1, CD68, P2RY12, TMEM119). No frozen tissue was collected for the study presented. We agree that, had we collected frozen tissue, a more comprehensive assessment of markers involved in neuronal development, as well as advanced quantification of microglia and their activation markers, could have been conducted.
- Even when drawing conclusions based solely on IHC, it is necessary to demonstrate semi-quantitative analysis (signal intensity measurement per ROI, cell counting, standardization procedures between slides, antibody validation data = blocking peptide and negative controls). Thank you. We have changed the references to our immunohistochemistry results to reflect their semi-quantitative nature, as suggested by the reviewer.
Added to limitations
“Consequently, the outcomes of the current study are based, in part, on semi-quantitative measures.”
- The authors counter that “previous studies in sheep demonstrate detectability using IHC alone,” but the cited literature does not guarantee the validity of IHC in this study. It is necessary to explicitly state whether experimental conditions such as antibodies, fixation conditions, and developmental stages are consistent. Yes, all experimental conditions were kept consistent within each IHC assessment, including antibodies, fixation protocol, tissue processing, and developmental stage, ensuring comparability across samples. We have added the following to the methods
“For all immunofluorescence analyses, experimental conditions were standardised within each assessment: tissues were collected at the same developmental stage, fixed and processed using identical protocols, and stained with the same antibody in the same run. Negative controls and blocking peptides were included to confirm specificity.”
